# Pectin/Activated Carbon-Based Porous Microsphere for Pb^2+^ Adsorption: Characterization and Adsorption Behaviour

**DOI:** 10.3390/polym13152453

**Published:** 2021-07-26

**Authors:** Ri-si Wang, Ya Li, Xi-xiang Shuai, Rui-hong Liang, Jun Chen, Cheng-mei Liu

**Affiliations:** 1State Key Laboratory of Food Science and Technology, Nanchang University, Nanchang 330047, China; ncuskwangrisi@163.com (R.-s.W.); shuaixixiang1989@163.com (X.-x.S.); liangruihong@ncu.edu.cn (R.-h.L.); 2South Subtropical Crops Research Institute, Chinese Academy of Tropical Agricultural Sciences, Zhanjiang 524091, China; liya1995ncu@163.com

**Keywords:** pectin, activated carbon, microsphere, Pb^2+^, adsorption

## Abstract

The development of effective heavy metal adsorbents has always been the goal of environmentalists. Pectin/activated carbon microspheres (P/ACs) were prepared through simple gelation without chemical crosslinking and utilized for adsorption of Pb^2+^. Scanning electron microscopy (SEM) revealed that the addition of activated carbon increased the porosity of the microsphere. Texture profile analysis showed good mechanical strength of P/ACs compared with original pectin microspheres. Kinetic studies found that the adsorption process followed a pseudo-second-order model, and the adsorption rate was controlled by film diffusion. Adsorption isotherms were described well by a Langmuir isotherm model, and the maximum adsorption capacity was estimated to be 279.33 mg/g. The P/ACs with the highest activated carbon (P/AC_2:3_) maintained a removal rate over 95.5% after 10 adsorption/desorption cycles. SEM-energy-dispersive X-ray spectrum and XPS analysis suggested a potential mechanism of adsorption are ion exchange between Pb^2+^ and Ca^2+^, electronic adsorption, formation of complexes, and physical adsorption of P/ACs. All the above results indicated the P/ACs may be a good candidate for the adsorption of Pb^2+^.

## 1. Introduction

With the development of rapid urbanization and industrialization, water pollution has become one of the major concerns of environmentalists and scientists around the world [1]. Among various water pollutants, heavy metals have drawn much attention due to their toxic characteristics. After they are enriched in aquatic organisms, they will be transferred step by step along the food chain and finally accumulate in the human body through food [2]. Lead is one of the most serious heavy metals and has a serious impact on the human body, causing damage to the hematopoiesis, kidney, and the nervous system [3]. Therefore, effectively removing Pb^2+^ from water has become an urgent problem around the world.

At present, several common wastewater treatment techniques include physicochemical and biological methods such as chemical precipitation, coagulation-flocculation, membrane separation, ion exchange, and physical adsorption [4]. Among the above methods, the adsorption method is one of the most effective techniques because of its high efficiency, low cost, environmental friendliness, and universality [5].

An effective method for the adsorption of Pb^2+^ is to chelate the Pb^2+^ with various functional groups such as carboxyl, hydroxyls, and phosphates. Natural polymer materials, such as pectin [6], lignin [7], and cellulose [8], have attracted wide attention in Pb^2+^ adsorption because of their biocompatibility and biodegradability. Pectin, a nontoxic and biocompatible complex anionic polysaccharide rich in carboxyl and hydroxyl groups, is considered to be a very promising adsorption material for wastewater treatment [9]. However, like other single biopolymer materials, pectin also has obvious drawbacks such as low surface area, difficulty in separation and regeneration, and poor mechanical strength [10], which makes it hard to meet the needs of practical applications [11]. Therefore, functional and/or physical modification has become critical.

Activated carbon, a promising adsorbent for heavy metal adsorption, is a carbonaceous material produced by various thermal decomposition methods [12]. The presence of diverse functional groups, highly porous structure, and relatively high mechanical strength are the important properties of activated carbons that influence the adsorption of heavy metals in wastewater [13]. However, powder-like activated carbon has the problems of easy loss and difficulty separating it from the solution after adsorption. Moreover, that which remains in the environment will be desorbed under specific conditions, which will cause secondary environmental pollution [14].

Although many papers have reported some effective porous materials for adsorption of Pb^2+^, the preparation of porous materials is often carried out under some harsh conditions such as high-temperature [15], tedious, and toxic chemical reactions [16,17]. For the purpose of improving the adsorption properties, mechanical strength, and easy separation of adsorbents from wastewater, we focused on a green synthesis of microsphere based on pectin and activated carbon, in which the composite may further benefit from the synergistic effects. In this work, pectin-based microspheres were prepared by adding different proportions of active carbon as a special functional material to obtain better properties for Pb^2+^ adsorption. The microspheres were characterized by Fourier transform infrared spectroscopy (FTIR), scanning electron microscopy (SEM), and surface area and pore size analysis, as well as texture profile analysis (TPA). The effects of pH, contact time, initial Pb^2+^ concentration on the adsorption of Pb^2+^, and reuse of adsorbents were tested. Moreover, the potential adsorption mechanism of adsorbents was investigated by energy-dispersive X-ray spectrum (EDX) and X-ray photoelectron spectroscopy (XPS).

## 2. Materials and Methods

### 2.1. Materials

Citrus pectin (type 104, molecular weight of 786 kDa, degree of methoxylation of 28.3%, degree of amidation of 20.63%) was supplied by the general agent of CPkelco (Shanghai, China). Activated carbon (AC), anhydrous calcium chloride, sodium hydroxide, hydrochloric acid, nitric acid (HNO_3_), and Pb(NO_3_)_2_ were purchased from Aladdin Reagent Company (Shanghai, China). All other chemicals and reagents used were of analytical reagent grade. All aqueous solutions were prepared with ultrapure water (18.2 MΩ cm resistance) from a Milli-Q system (Millipore, Billerica, MA, USA).

### 2.2. Preparation of Pectin/Active Carbon Microspheres

Firstly, citrus pectin (2.0 g) was dissolved in 98 mL deionized water with magnetic stirring at room temperature for 12 h. Activated carbon was added at a mass ratio of pectin to activated carbon of 4:1, 2:1, 1:1, and 2:3, and then continuous mechanical stirring at room temperature for 3 h to obtain a homogeneous mixture. The pectin/activated carbon mixture was dripped into a Petri dish containing about 100 mL of 0.5 mol/L calcium chloride solution through a syringe without needle (stirring continuously at the bottom). The pectin/activated carbon microspheres (P/ACs) were rinsed off consecutively with deionized water several times and dried at 30 °C for 12 h (10–20 undried hydrogel microspheres were retained for texture analysis). The resulting products were named P/AC_4:1_, P/AC_2:1_, P/AC_1:1_, and P/AC_2:3_ respectively. Pectin microspheres without added activated carbon were prepared as a control group.

### 2.3. FTIR Spectra Analysis

Fourier transform infrared spectra of adsorbents were measured by Nicolet 5700 FTIR (Thermo Fisher Scientific, Madison, WI, USA). A 1.0 mg sample and KBr were mixed and pressed in the mass ratio of 1:100. The spectral analysis was carried out at the scanning frequency of 4000 to 400 cm^−1^. Data were collected and analyzed by Ominic 7.2 software [18].

### 2.4. SEM Analysis

The dried samples were adhered to a circular specimen stub with conductive adhesive. The external morphology of samples was measured by SEM (model JSM-6701F, JEOL, Japan) under different magnifications [19].

### 2.5. Specific Surface Area and Pore Size Analysis

The samples were placed on a specific surface area and pore size analyzer (JW-BK132 F, JWGB, China) and analyzed by the nitrogen adsorption at 77 K. Specific surface area was calculated by Brunauer–Emmett–Teller (BET) algorithm, and pore volume and average pore size were calculated by the Barrett–Joyner–Halenda (BJH) method [20].

### 2.6. Texture Profile Analysis

The mechanical strength of the sample was measured using a texture analyzer (TA-XT plus, SMS Ltd., London, UK) at room temperature. The microspheres were submitted to texture profile analysis (TPA) using a probe P36R, and the following test parameters were used: constant speed of 1.0 mm/s, trigger force of 5.0 g, and compression of 50%.

### 2.7. Batch Adsorption Experiments

A 1.589 g sample of Pb(NO_3_)_2_ was dissolved in 1.0 L of distilled water. Different concentrations of Pb(NO_3_)_2_ solution were obtained by diluting the stock solution (1.0 g/L) of Pb^2+^. In order to understand the effect of pH, contact time, and initial Pb^2+^ concentration on adsorption of Pb^2+^, pH was set in the range of 2.0–6.0, contact time of 10 min to 1440 min, and the initial Pb^2+^ concentration of 20 mg/L to 300 mg/L. After adsorption, the microspheres were separated from the solutions using a filter. The initial and final Pb^2+^ concentrations were determined by an atomic absorption spectrophotometer (AAS, A3AFG-12, Puxi, Beijing, China). The removal rate (R) and the adsorption capacity (q_e_) for Pb^2+^ are calculated as follows:(1)$$R=\frac{\;C_{0}{-C}_{e}}{C_{0}}\;\times \;100$$
(2)$$q_{e}=\frac{{V(C}_{0}{-C}_{e})}{m}\;\times \;100$$
where C_0_ and C_e_ (mg/L) are the initial and equilibrium concentrations of Pb^2+^ solution, respectively. V (L) is the volume of the solution, and m (g) is the mass of the adsorbent.

The experimental results of contact time and initial Pb^2+^ concentration were used for adsorption kinetics and isotherm study, respectively (Table 1).

### 2.8. Adsorption–Desorption Cycles

Adsorbent was added into Pb^2+^ solution with an initial concentration of 50 mg/L, adsorbent dosage of 1.0 g/L, and pH of 5.0. After adsorption of 1440 min, adsorbents were filtrated, and the concentration of Pb^2+^ in the filtrate was determined by atomic absorption spectrophotometer. After that, adsorbents were added into 30 mL 0.1 mol/L HNO_3_ solution and stirred for 30 min. The adsorbent was then filtered and washed with distilled water several times. This adsorption–desorption was repeated 10 times.

### 2.9. SEM-EDX Analysis

The surface of samples before and after adsorption processes were analyzed with the combination of Zeiss Ultra Plus SEM (Carl Zeiss NTS, Jena, Germany) and XFlash Quad 5060F EDX (Bruker Nano GmbH, Berlin, Germany).

### 2.10. X-ray Photoelectron Spectroscopy Analysis

The surface chemical composition of samples before and after adsorption was analyzed by X-ray photoelectron spectroscopy (ESCALAB250Xi, Thermo Fisher Scientific, Madison, WI, USA). The test was excited by AlKα ray and corrected by C1s (284.8 ev). The obtained map was processed by XPS PEAK4.1 software [23].

## 3. Results and Discussion

### 3.1. Characterization of P/ACs

The functional groups of microspheres were detected by FTIR and are shown in Figure 1. There is a broad band of 3395.5 cm^−1^ that appeared in the spectrum of pectin microspheres, which was assigned to stretching vibration of O-H groups. Two peaks at about 1627 cm^−1^ and 1441 cm^−1^ exhibited the vibrations of the carboxylate group (-COO) [24]. The typical bands of activated carbon were viewed at 1575.1 cm^−1^ assigned to the C=O peak of ketone or carbonyl, and the 1212.1 cm^−1^ was the C-O peak of aliphatic ether or phenol [25]. It was found that the peak around 3395.5 cm^−1^ gradually disappeared, and peaks of 1627 cm^−1^ and 1441 cm^−1^ shifted to lower wavenumber with the increase of activated carbon content. The spectrum of P/ACs seems to be the combination of pectin and activated carbon, indicating that the preparation of P/ACs was mainly a physical complexing process without changing functional groups of both pectin and activated carbon.

The surface morphologies of the pectin microsphere and P/ACs were observed by SEM analysis and are shown in Figure 2. The surface morphology of the original pectin microsphere exhibited a slightly wrinkled surface without pores or cracks on it, which was quite similar to the morphology of the pectin microsphere reported by other researchers [26]. In comparison, the size of microspheres seemed to increase with the addition of activated carbon. The size of pectin microsphere, P/AC_4:1_, P/AC_2:1_, P/AC_1:1_, and P/AC_2:3_ was 1.30 ± 0.18 mm, 1.44 ± 0.15 mm, 1.75 ± 0.17 mm, 2.28 ± 0.19 mm, and 2.78 ± 0.20 mm, respectively. Meanwhile, the surface of P/ACs was seen to be more coarse. With the increased amount of activated carbon, more and more pores and cavities were found, indicating that the constancy of pectin gel was interrupted. The structures of these cavities may make good opportunities for trapping and adsorbing the Pb^2+^ onto the exterior of the P/ACs.

The N_2_-adsorption isotherms and pore volume distributions are presented in Figure 3. The data of the specific surface area (SSA), average pore size (PS), and pore volume (PV) of the obtained samples are listed in Figure 3F. It can be observed that the P/ACs showed Type Ⅳ isotherms and H4 hysteresis loop, which can be attributed to the presence of microporous and mesopores [27]. In addition, it was found that the SSA and PV of P/ACs were dramatically increased with the addition of activated carbon, especially for those containing a high number of activated carbons (P/AC_1:1_ and P/AC_2:3_). Generally, the large SSA could provide a large number of available active sites and hierarchical structure, which endow the adsorbent with excellent adsorption performance [28]. In addition, larger PV and smaller average PS indicated larger porosity of the adsorbent, which may be more beneficial for Pb^2+^ to bind with potential active sites.

The mechanical properties of microspheres were studied through TPA compression tests. It was found that the hardness of microspheres increased with the number of activated carbons, and the P/AC_2:3_ showed a gel hardness of 181.41 g, which was greater than that of the pectin microsphere (128.52 g), P/AC_4:1_ (140.61 g), P/AC_2:1_ (150.31 g), and P/AC_1:1_ (154.20 g). These results suggest that the addition of activated carbon played an important role in improving the mechanical strength of the microsphere matrix. In addition, good mechanical strength not only means that the microspheres can withstand the washing and stirring of mechanical force in the actual wastewater treatment, but also means that the microspheres can bear greater column pressure when they are used as fillers.

Based on the above results, it is expected that the P/ACs will provide greater removal efficiency of Pb^2+^ as compared to the original pectin microspheres due to their favorable porosity and mechanical strength.

### 3.2. Effect of Initial pH on the Pb^2+^ Adsorption

Solution pH is one of the most important factors determining the surface charge along with significant roles in the adsorption of target metal ions. From Figure 4, it is clear that at low pH (pH = 2.0), all the samples showed relatively poor adsorption efficiency. With the increase in pH from 2.0 to 4.0, the removal rate increased rapidly, and the optimum pH range of 5.0–6.0 was obtained for all samples studied. This can be explained by the pKa of pectin being about 3.5–4.1. When pH value is lower than pKa, the protonation of carboxyl sites of pectin molecule was favored, and there is a competitive relationship between the positively charged H^+^ and Pb^2+^, leading to a reduced number of COO^−^ for Pb^2+^ binding, so the adsorption amount is small. When pH values became higher than pKa, the majority of carboxyl groups were in a dissociated state, and there were more and more deprotoned carboxylic acid groups and free hydroxyl groups available in the solution. These groups can be used to adsorb Pb^2+^, thus improving the adsorption efficiency [29].

### 3.3. The Kinetic Studies

The effect of contact time on the removal rate of Pb^2+^ was investigated, and the results are shown in Figure 5A. It was obvious that the uptake of Pb^2+^ was very fast in the first 2 h. This rapid adsorption could be attributed to the sufficient unused adsorption sites and the large concentration gradient, which enables the rapid adsorption of Pb^2+^ to the surface of the adsorbent. As the active site on the adsorbent surface was gradually occupied by Pb^2+^, the repulsive force on the solid–liquid interface gradually increased, leading to a decrease in the adsorption rate of Pb^2+^ until it reached equilibrium.

Kinetic studies are fundamental in adsorption science and technology. In order to understand the kinetic behavior of Pb^2+^ adsorbed by the P/ACs, three kinetic models, including the pseudo-first-order, pseudo-second-order, and intra-particle diffusion, were investigated. The linear equations of these models are compiled in Table 1, and their linear fitting plots and parameters are shown in Figure 5B–D and Table 2. The pseudo-second-order model fitted the experimental data better, with relatively high coefficients of determination (*R*^2^ > 0.996), as compared with the pseudo-first-order model. According to the pseudo-second-order model, the saturation adsorption capacity (q_e_) of P/AC_2:3_ was 50.97 mg·g^−1^, which was quite consistent with the experimental value (48.98 mg·g^−1^). According to the principle of pseudo-second-order model, it was inferred that the adsorption of Pb^2+^ by P/ACs was dominated by chemisorption, similar to the results of Chwastowski et al. [30]. The chemisorption such as some electron transfer or sharing between the Pb^2+^ and the adsorbents may be the rate-limiting step [31]. In addition, when the experimental results were fitted into the intraparticle diffusion, the plot exhibits a multi-linearity relationship (Figure 5D). The first sharper linear curve represents the surface adsorption, whereas the second one corresponds to the slow diffusion of Pb^2+^ from the surface to the inner pores. The third linear portion is approaching equilibrium. For P/AC_2:3_, the external mass transfer rate constant (Kp_1_ = 3.212 mg/g min^1/2^) is much greater than the internal surface diffusion rate constant (Kp_2_ = 0.779 mg/g min^1/2^) and equilibrium diffusion rate constant (Kp_3_ = 0.043 mg/g min^1/2^), suggesting that the overall adsorption rate is mainly controlled by film diffusion as the primary rate-determining step.

### 3.4. The Adsorption Isotherms Study

An adsorption isotherm study was carried out with initial Pb^2+^ concentrations between 20 mg/L and 300 mg/L. As shown in Figure 6A,B, the adsorption capacities of all the samples increased with the initial Pb^2+^ concentration, which may be because the initial heavy metal concentration gradient is the main driving force for mass transfer of heavy metal from the solution to the adsorbent [32]. However, the removal rate (%) decreased with initial Pb^2+^ concentration, which may because the number of available adsorption sites was limited for a certain amount of adsorbent. When the binding site of the adsorbent was occupied, even the concentration of Pb^2+^ is increased, its adsorption capacity did not increase much, showing a decrease in the removal rate. In addition, it was found that no significant difference in adsorption capacities between the original pectin microsphere and P/ACs at low Pb^2+^ concentration (20 mg/L and 40 mg/L), but the adsorption capacities of P/ACs were much higher than those of the original pectin microsphere at high Pb^2+^ concentration. For example, the P/AC_2:3_ kept more than 80% removal rate when the initial Pb^2+^ concentration reached 100 mg/L, and the removal rate of P/AC_2:3_ was 1.72 fold of pectin microsphere when initial Pb^2+^ concentration reached 300 mg/L, indicating supplementary activated carbon significantly increased adsorption capacity of pectin microsphere.

Langmuir, Freundlich isotherm models are used for modeling experimental data of Pb^2+^ adsorption on P/ACs. The linear equations of these models are listed in Table 1. Figure 6C–D shows the fitting results of adsorption isotherm, and data are summarized in Table 3. It was found that the correlation coefficients (*R*^2^) of the Langmuir model (>0.994) are bigger than the *R*^2^ of the Freundlich model (0.957–0.983). Therefore, adsorption data fitted well with the Langmuir model than the Freundlich model. Considering that the Langmuir isotherm assumes that all adsorption sites are similar and do not affect each other [33], the adsorption of Pb^2+^ onto the P/ACs was attributed to energetically homogeneous binding sites on the P/ACs surface. According to the Langmuir fitting, the theoretical maximum uptake (Q_m_) of Pb^2+^ was 279.33 mg/g. This value is greater than that of many other polysaccharide-based adsorbents reported recently [34,35]. The effective adsorption of P/ACs towards Pb^2+^ might be attributed to the introduction of activated carbon. The Q_m_ of activated carbon was found to be 256.3 mg/g. This value was significantly higher than that of the pectin microsphere (120 mg/g), which is why increasing the content of activated carbon in the microsphere increased the adsorption results. In addition, the higher specific surface area of P/ACs would offer more active binding sites and stronger adsorbent–adsorbate interactions for adsorption.

### 3.5. Desorption and Regeneration

The regeneration of adsorbent is one of the important indexes for assessing its practical applications. Using 0.1 mol/L HNO_3_ as an eluent helps to rerelease metal ions from the binding sites of the adsorbent. At lower pH values, the inhibition of Pb^2+^ adsorption also indirectly supports this statement. The performances of P/ACs during ten consecutive desorption regeneration cycles are shown in Figure 7. The results showed that the adsorption performance of the pectin microsphere became worse and worse during adsorption-desorption cycles. The pectin microsphere was gradually dissolved into a solution so that almost no microsphere could be filtrated after ten cycles. However, it was found that there was little decrease (4.5%) in adsorption performance after ten cycles of P/AC_2:3_. These results indicate that P/ACs possess excellent reusability for the effective removal of Pb^2+^ from aqueous media, which may be partly attributed to the high mechanical stability.

### 3.6. Adsorption Mechanism

In order to study the adsorption mechanism of P/ACs towards Pb^2+^, EDX element scanning analysis was performed on the surface of P/ACs before and after adsorption (Figure 8). Before adsorption, samples showed the existence of carbon, oxygen, chloride, and calcium elements on the surfaces. After adsorption, the EDX results confirmed the successful adsorption of Pb^2+^ on the P/ACs surface. Moreover, after adsorption, the content of calcium and chloride element in microspheres decreased significantly, which may be caused by the ion exchange between Ca^2+^ and Pb^2+^ during the adsorption process.

XPS analysis was performed to understand the interactions between the adsorbent and Pb^2+^ based on the involved binding energy. XPS spectra of samples before and after Pb^2+^ adsorption were implemented in Figure 9. The wide scan XPS spectra of Pb4f (Figure 9B) showed two strong peaks appearing at 144.1 eV and 139.1 eV, which were assigned to Pb4f_5/2_ and Pb4f_7/2_, respectively. The newly detected Pb4f peak confirmed the adsorption of Pb^2+^ on the surface of the adsorbent. Interestingly, the Ca2s and Cl2s peak (Figure 9A) could be seen in the spectrum before adsorption, but its strength almost disappeared after adsorption. This may indicate that there was ion exchange between Pb^2+^ and Ca^2+^ in the adsorption process; this result was consistent with EDX results. It is well known that the surface functional groups, such as -COOH group and -OH group, had an excellent Pb^2+^ adsorption function due to the electron donor to the cationic. The C1s spectra of P/AC_2:3_ before and after heavy metals adsorption can be deconvoluted to three peaks (Figure 9C–D). The binding energy for the carbon atoms in the -C-O-C-, -C-OH, and -O-C=O were increased and the peaks shifted from 286.2 eV to 286.5 eV, 287.2 eV to 287.3 eV, and 289.0 eV to 289.2 eV, respectively, indicating that the three functional groups were involved in combination with heavy metals [36]. It can also be observed from the O1s (Figure 9E,F) spectrum that two peaks lying at 533.90 eV and 532.20 eV correspond to -C-O- and -O-C=O. After Pb^2+^ adsorption, the peak shifted to 533.5 eV and 531.60 eV, indicating that these groups may complex with Pb^2+^ [37].

According to the results of isotherm studies, SEM-EDX, and XPS, the adsorption of Pb^2+^ by P/ACs may involve ion exchange, electronic adsorption, complexing, and physical adsorption [38], as illustrated in Figure 10.

## 4. Conclusions

Pectin/activated carbon microspheres (P/ACs) with highly porous structures and excellent mechanical strength were successfully prepared through a simple combination without chemical crosslinking. Batch experiments demonstrated that the adsorption of P/ACs was dominated by chemisorption, and the adsorption rate is simultaneously controlled by film diffusion. EDX imaging, as well as XPS analysis, indicated that P/ACs removed Pb^2+^ through ion exchange, complexation, and physical adsorption. After 10 adsorption–desorption cycles, the removal rate of P/AC_2:3_ was maintained at 95.5%. These results indicated that P/AC_2:3_ was an excellent and eco-friendly adsorbent for the removal of Pb^2+^ from wastewater.

## Figures and Tables

**Figure 1 polymers-13-02453-f001:**
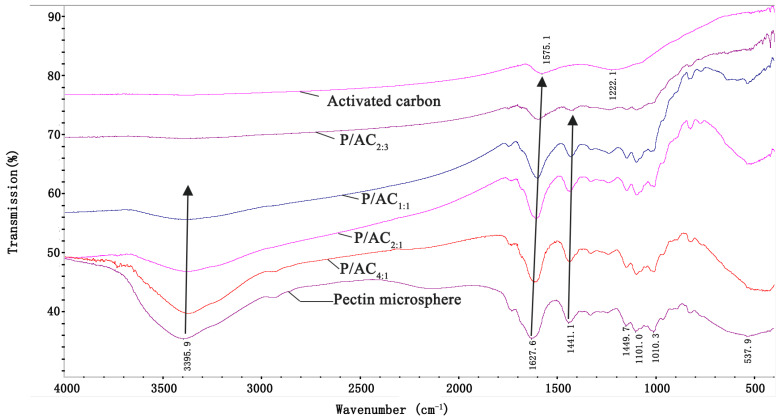
FTIR spectra of pectin microsphere, activated carbon, and P/ACs.

**Figure 2 polymers-13-02453-f002:**
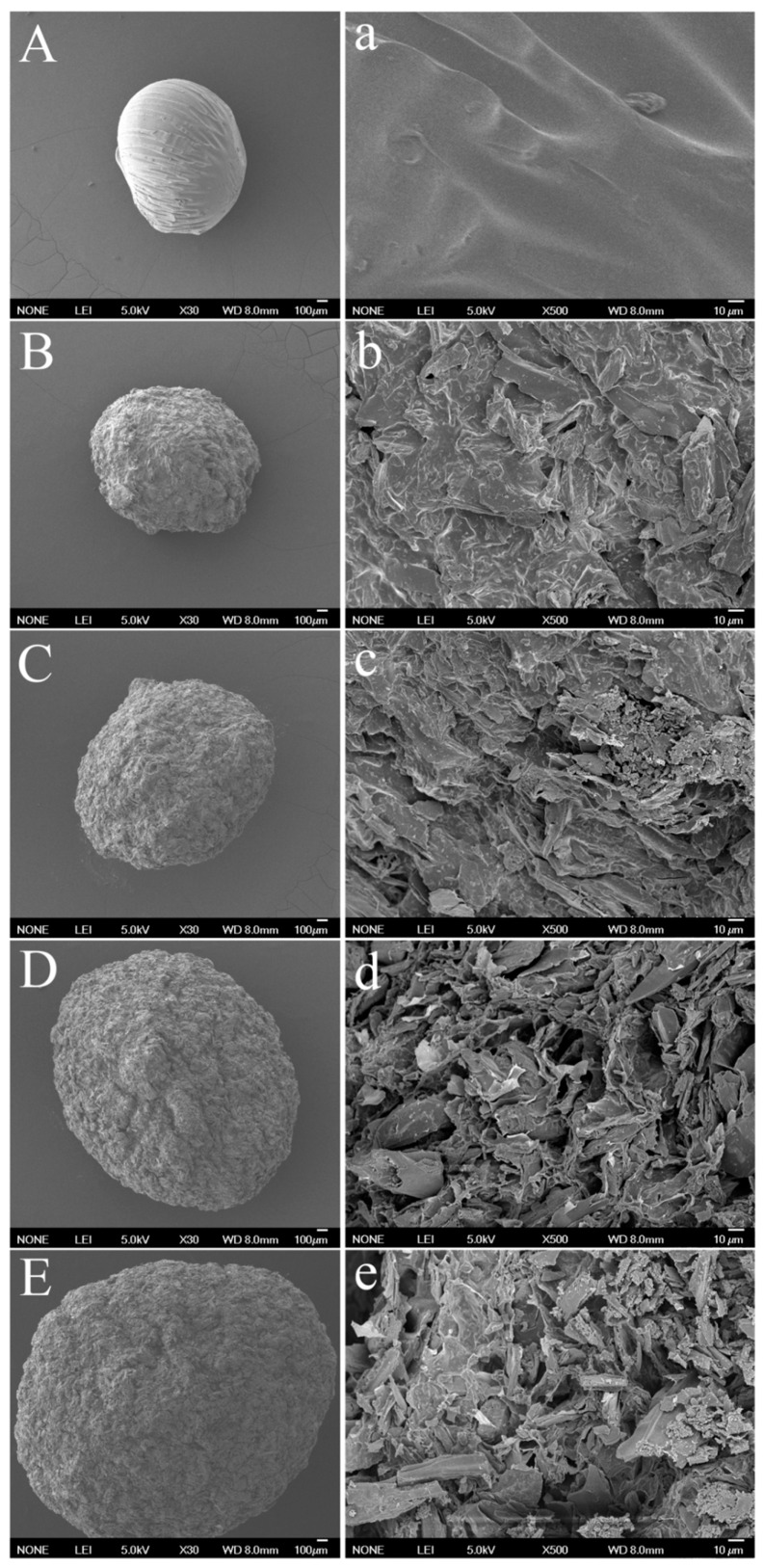
SEM images of pectin microsphere (**A**,**a**), P/AC_4:1_ (**B**,**b**), P/AC_2:1_(**C**,**c**), P/AC_1:1_(**D**,**d**), and P/AC_2:3_(**E**,**e**). The capital letters represent the whole view of microspheres, and the lower-case letter represents the enlarged surface of microspheres.

**Figure 3 polymers-13-02453-f003:**
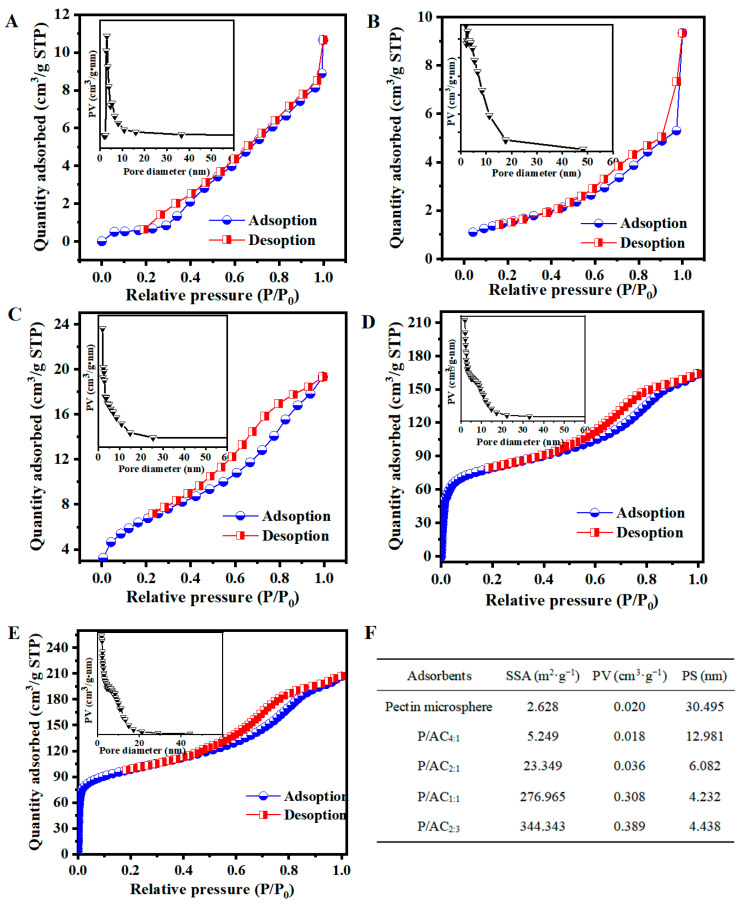
Nitrogen adsorption/desorption isotherms of (**A**) pectin microsphere, (**B**) P/AC_4:1_, (**C**) P/AC_2:1_, (**D**) P/AC_1:1_, and (**E**) P/AC_2:3_ (Inset: Pore-volume distribution). (**F**) the specific surface area (SSA), average pore size (PS), and pore volume (PV) of pectin microsphere and P/ACs.

**Figure 4 polymers-13-02453-f004:**
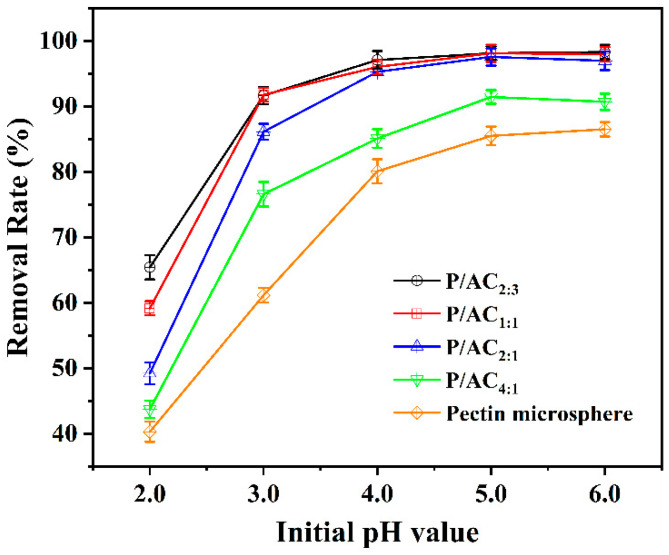
The effect of initial pH (2.0, 3.0, 4.0, 5.0, 6.0) on Pb^2+^ adsorption efficiency of P/ACs (adsorbent dosage: 1.0 g/L, initial Pb^2+^ concentration of 50 mg/L, temperature of 298 K, contact time of 1440 min).

**Figure 5 polymers-13-02453-f005:**
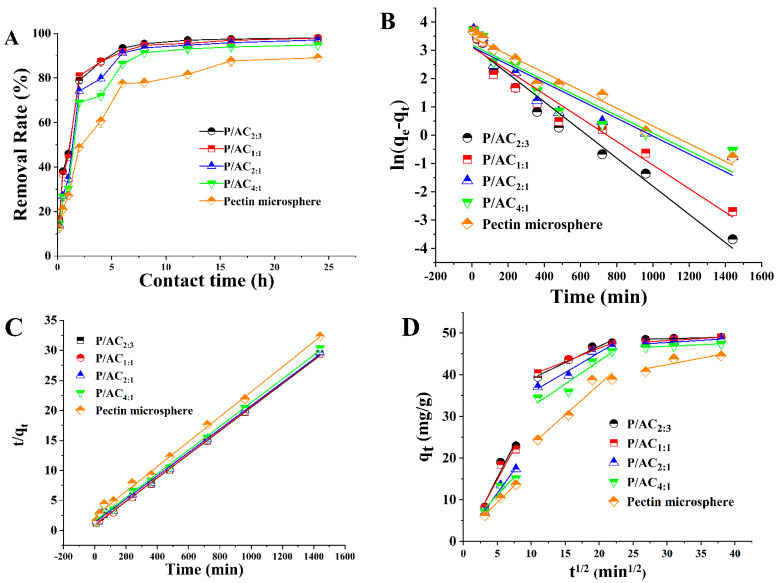
The effect of contact time (10, 30, 60, 240, 360, 720, 960, and 1440 min) on the removal rate of Pb^2+^ adsorbed by P/ACs (**A**). The pseudo-first-order kinetic model (**B**), the pseudo-second-order kinetic model (**C**), and the intraparticle diffusion plots (**D**) for Pb^2+^ adsorpted by P/ACs (adsorbent dosage of 1.0 g/L, initial Pb^2+^ concentration of 50 mg/L, temperature of 298 K, pH of 5.0).

**Figure 6 polymers-13-02453-f006:**
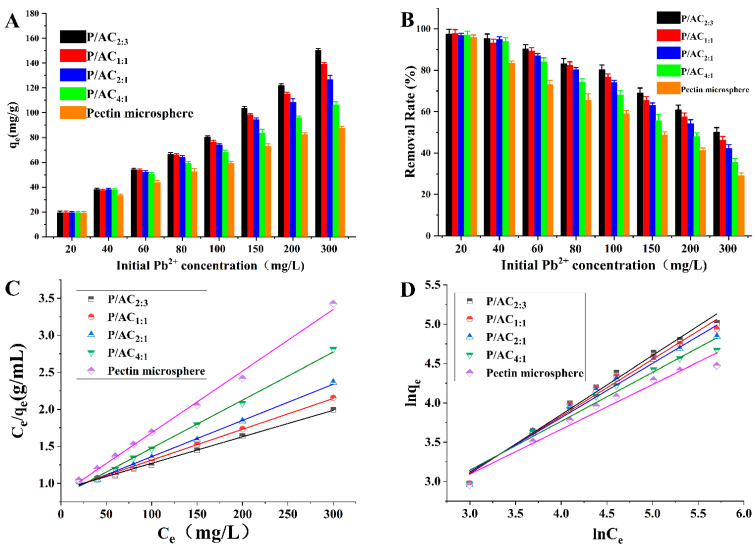
The effect of initial Pb^2+^ concentration (20, 40, 60, 80, 100, 150, 200, and 300 mg/L) on adsorption capacity (**A**) and removal rate (**B**) of P/ACs towards Pb^2+^. Adsorption isotherms for Pb^2+^ according to Langmuir model (**C**) and Freundlich model (**D**) (adsorbent dosage of 1.0 g/L, solution pH of 5.0, temperature of 298 K, contact time of 1440 min).

**Figure 7 polymers-13-02453-f007:**
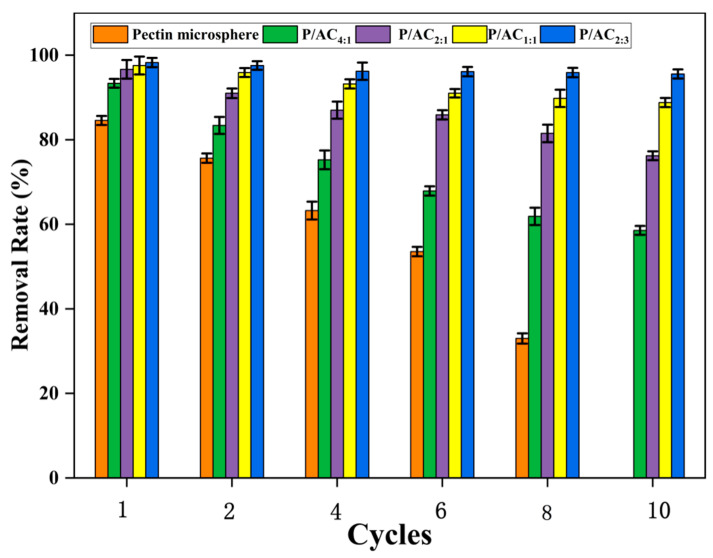
Efficiency of P/ACs on removal Pb^2+^
during adsorption–desorption cycles.

**Figure 8 polymers-13-02453-f008:**
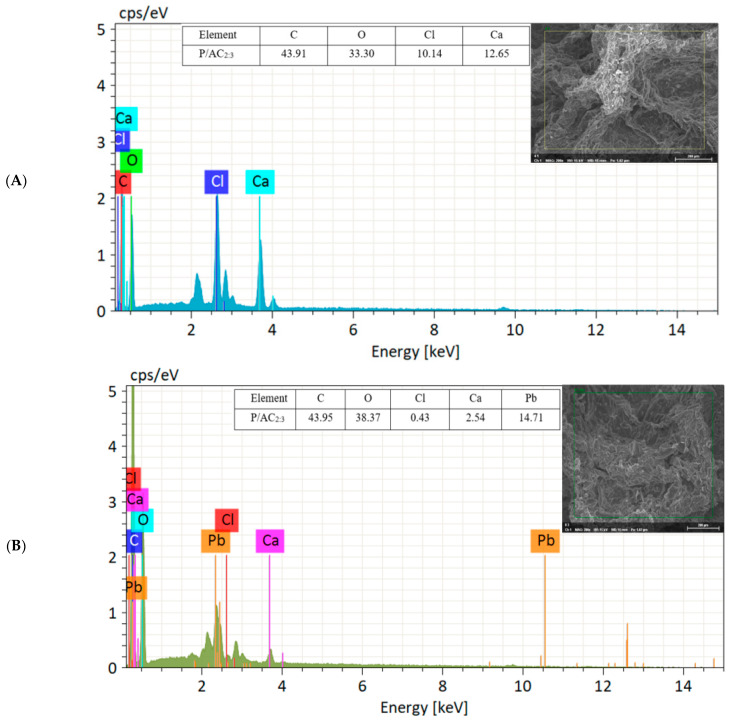
EDX image of P/AC_2:3_ before (**A**) and after (**B**) Pb^2+^ adsorption.

**Figure 9 polymers-13-02453-f009:**
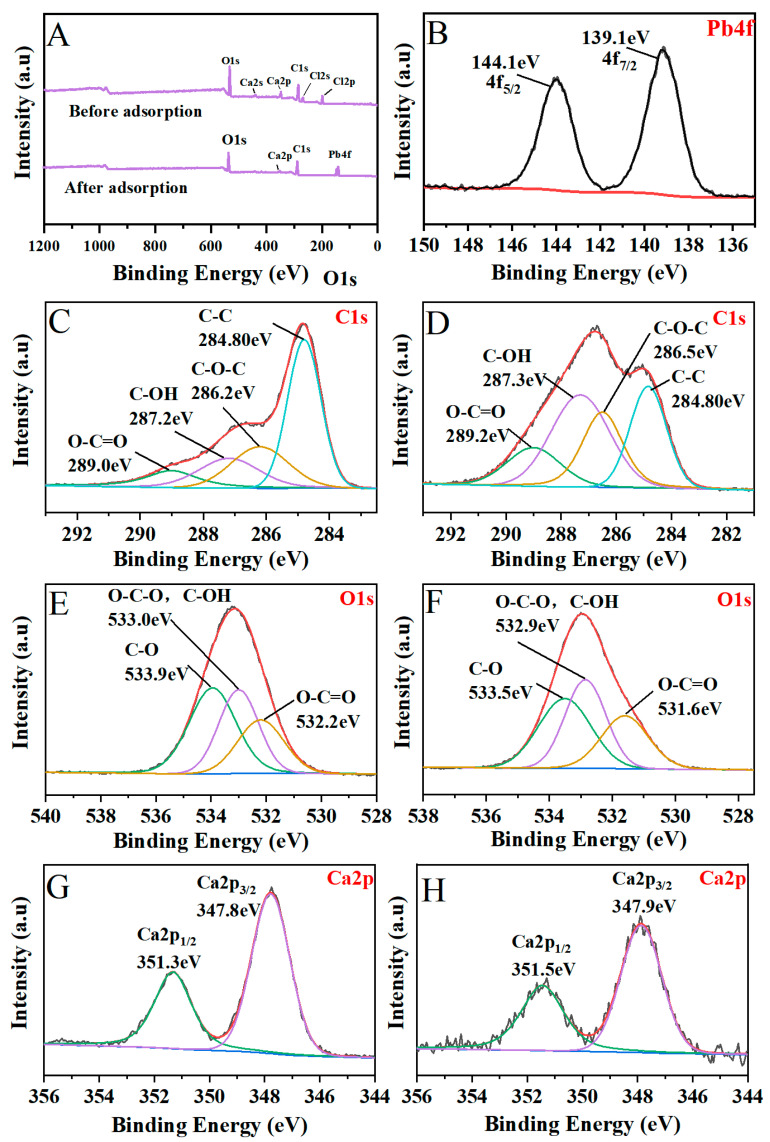
Wide-scan XPS spectra of P/AC_2:3_ before and after adsorption (**A**). High-resolution XPS spectra of Pb4f of P/AC_2:3_ after adsorption (**B**). O1s before (**C**) and after (**D**) adsorption, C1s before (**E**) and after (**F**) adsorption. Ca2p before (**G**) and after (**H**) adsorption.

**Figure 10 polymers-13-02453-f010:**
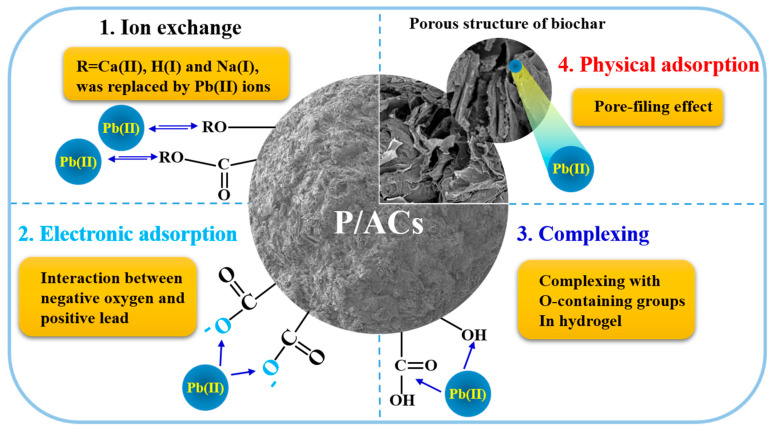
The proposed mechanisms of adsorption of Pb^2+^ by P/ACs.

**Table 1 polymers-13-02453-t001:** Kinetics and adsorption isotherm models used to study the adsorption process.

Model	Linear Equation	Parameters	Reference
Pseudo-first-order	$\ln\left(q_{e}{-q}_{t}\right){=\mathrm{lnq}}_{e}-k_{1}t$	$q_{t}$(mg/g): amounts of Pb^2+^ adsorbed at time *t*	[20,21]
		$q_{e}$(mg/g): amounts of Pb^2+^ adsorbed at time *t* at adsorption equilibrium	
		$k_{1}$(min^−1^): rate constants of the pseudo-first-order	
Pseudo-second-order	$\frac{t}{q_{t}}=\frac{1}{k_{2}q_{e}}+\frac{t}{q_{e}}$	$k_{2}$(min^−1^): rate constants of the pseudo-second-order	
Intra-particle diffusion	$q_{t}{=k}_{p}t^{1/2}+C$	$k_{p}$(mg·g^−1^·min^−1/2^): rate constant of intra-particle diffusion	
		C: intercept	
Langmuir	$\frac{1\;}{q_{e}\;}=\frac{1}{q_{m}}+\frac{1}{K_{L}q_{m}C_{e}}$	$q_{e}$(mg/g): Pb^2+^ adsorption at the equilibrium	[22]
		$C_{e}$(mg/L): equilibrium concentration of the Pb^2+^	
		$K_{L}$(L/mg): Langmuir constant	
		$q_{m}$(mg/g): theoretical maximum adsorption capacity of adsorbent	
Freundlich	lnq_e_= lnK_F_ + 1/$n_{F}$lnC_e_	$K_{F}$: Freundlich constant	
		$n_{F}$: intensity of the adsorbents	

**Table 2 polymers-13-02453-t002:** Kinetic model parameters of the adsorption of Pb^2+^ by P/ACs.

Adsorbent	q_e_, _exp_ (mg·g^−1^)	Pseudo-First-Order Model			Pseudo-Second-Order Model			Intra-Particle Diffusion Model		
		q_e_, _cal_ (mg·g^−1^)	K_1_ (min^−1^)	*R* ^2^	q_e_, _cal_ (mg·g^−1^)	K_2_ (g·mg^−1^·min^−1^)	*R* ^2^	Kp_1_	Kp_2_	Kp_3_
								(mg·g^−1^·min^−1/2^)		
Pectin microsphere	44.55	31.99	0.003	0.969	48.15	0.0089	0.997	1.619	1.454	0.312
P/AC_4:1_	47.49	24.42	0.003	0.879	50.66	0.0126	0.996	1.746	1.101	0.078
P/AC_2:1_	48.54	22.74	0.003	0.873	51.39	0.0145	0.997	2.267	0.955	0.103
P/AC_1:1_	48.94	22.60	0.004	0.947	50.86	0.0209	0.999	3.034	0.638	0.101
P/AC_2:3_	48.98	23.75	0.005	0.968	50.97	0.0219	0.999	3.212	0.779	0.043

**Table 3 polymers-13-02453-t003:** Isotherm parameters, correlation coefficients for adsorption of Pb^2+^ by P/ACs, and original pectin microsphere.

Adsorbents	Q_m_	Langmuir		Freundlich	
	(mg/g)	K_L_ (L/mg)	*R* ^2^	1/n_F_	*R* ^2^
Pectin microsphere	120.19	0.009	0.994	0.569	0.959
P/AC_4:1_	154.32	0.008	0.995	0.622	0.957
P/AC_2:1_	204.08	0.006	0.995	0.689	0.970
P/AC_1: 1_	240.38	0.005	0.997	0.722	0.978
P/AC_2:3_	279.33	0.004	0.996	0.751	0.983

## Data Availability

The data presented in this study are available on request from the corresponding author.

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
