# Peer review of "Pectin/Activated Carbon-Based Porous Microsphere for Pb2+ Adsorption: Characterization and Adsorption Behaviour"

_polymers, 2021, doi:10.3390/polym13152453_

Round 1
Reviewer 1 Report
Authors should check the manuscript because sometimes they write absorption and sometimes adsorption which is a completely different phenomenon.
In the fig.6 caption authors made a mistake - B is the removal % rate and C is the Langmuir isotherm model.
Please kindly add citation:
https://doi.org/10.3390/ma13122782
Reviewer 2 Report
Journal: Polymers
Manuscript Number: polymers-1295072
Type of the Paper: Article
Title: Pectin/activated carbon based porous microsphere for Pb2+ adsorption: Characterization and adsorption behavior
The authors have described the preparation of pectin/activated carbon-based porous microsphere and the prepared microsphere was utilized for Pb2+ adsorption. The paper is well-written and organized. Thus, I suggest the manuscript for publication after considering the below points.
1) There are many reports available regarding the adsorption of heavy metal ions using porous materials. Thus, the novelty and advantages of this method should be present in the last part of the introduction section.
2) The results with increasing the activated carbon showed good results. What will be the adsorption rate using just activated carbon? In the paper adsorption using pectin are showed but there no information just will activated carbon. Also increasing the content of activated carbon in the microsphere increase the adsorption results.
3) The scale in the SEM images is not visible. Table or information should be added reading the size of the microsphere because the SEM images reveal the increase in size but there is no information about this.
4)The adsorption results in changes with pH values that need more explanation. Why the adsorption between 4.0 and 6.0 increase?
5) Why the removal rate (%) decreases with initial Pb2+ concentration?
